# Differences in Milk Proteomic Profiles between Estrous and Non-Estrous Dairy Cows

**DOI:** 10.3390/ani13182892

**Published:** 2023-09-12

**Authors:** Chao Du, Liangkang Nan, Chunfang Li, Chu Chu, Haitong Wang, Yikai Fan, Yabin Ma, Shujun Zhang

**Affiliations:** 1College of Animal Science and Veterinary Medicine, Henan Institute of Science and Technology, Xinxiang 453003, China; dc1992hml@163.com; 2Key Lab of Agricultural Animal Genetics, Breeding and Reproduction of Ministry of Education, Huazhong Agricultural University, Wuhan 430070, China; 15827557518@163.com (L.N.); chunfangli0521@126.com (C.L.); chu18506321907@163.com (C.C.); wanghaitong0411@163.com (H.W.); fanyikai123@126.com (Y.F.); 3Hebei Livestock Breeding Station, Shijiazhuang 050000, China

**Keywords:** dairy cows, milk, estrus, biomarkers, proteomics

## Abstract

**Simple Summary:**

Discovering estrus biomarkers is of utmost importance to develop an easy, reliable, and accurate estrus detection method in dairy cows. The proteomics approach is an attractive and powerful tool and recently has been used to explore proteomic alterations in a variety of bodily fluids to identify the relative biomarkers. Milk produced by dairy cows is a complex biofluid consisting of hundreds of different components that may be a good source for biomarker discovery. In this study, the proteomics approach was performed on milk sampled from estrus and non-estrus dairy cows to identify potential biomarkers of estrus, which may be helpful for further development of an easy and reliable estrus detection method in dairy cows.

**Abstract:**

Efficient reproductive management of dairy cows depends primarily upon accurate estrus identification. However, the currently available estrus detection methods, such as visual observation, are poor. Hence, there is an urgent need to discover novel biomarkers in non-invasive bodily fluids such as milk to reliably detect estrus status. Proteomics is an emerging and promising tool to identify biomarkers. In this study, the proteomics approach was performed on milk sampled from estrus and non-estrus dairy cows to identify potential biomarkers of estrus. Dairy cows were synchronized and timed for artificial insemination, and the cows with insemination leading to conception were considered to be in estrus at the day of insemination (day 0). Milk samples of day 0 (estrus group) and day −3 (non-estrus group) from dairy cows confirming to be pregnant were collected for proteomic analysis using the tandem mass tags (TMT) proteomics approach. A total of 89 differentially expressed proteins were identified, of which 33 were upregulated and 56 were downregulated in the estrus milk compared with the non-estrus milk. Gene Ontology (GO) and Kyoto Encyclopedia of Genes and Genomes (KEGG) enrichment analysis showed that acetyl coenzyme A carboxylase α (ACACA), apolipoprotein B (APOB), NAD(P)H steroid dehydrogenase-like (NSDHL), perilipin 2 (PLIN2), and paraoxonase 1 (PON1) participated in lipid binding, lipid storage, lipid localization, and lipid metabolic process, as well as fatty acid binding, fatty acid biosynthesis, and fatty acid metabolism, and these processes are well documented to be related to estrus regulation. These milk proteins are proposed as possible biomarkers of estrus in dairy cows. Further validation studies are required in a large population to determine their potential as estrus biomarkers.

## 1. Introduction

The precise identification of estrus is essential for efficient reproduction management in dairy cows. Estrus is traditionally detected through visual observation of some well characterized physiological and social behaviors, such as swelling and reddening of the vulva, restless, sniffing and licking the vulva of another cow, and standing to be mounted [1]. Nevertheless, the detection of estrus by visual observation is often inefficient because only 50–60% of cows express estrus signs. Moreover, visual observation is labor intensive, and the development of modern large dairy herds pattern hinders this method to a certain extent [2]. Nowadays, several automated technologies such as pressure-sensitive devices, activity monitoring devices, and body temperature detecting devices have been developed to detect estrus [3]. Nevertheless, the accuracy of these devices is still not completely reliable, and these technologies require potentially burdensome investments in management and equipment [4]. Therefore, discovering estrus biomarkers is of utmost importance for developing an easy, reliable, and accurate estrus detection method.

Proteomics is an attractive and powerful tool and recently has been used to explore proteomic alterations in a variety of bodily fluids to identify biomarkers related to estrus [5]. For instance, Zhao et al. [6] identified 14 differentially expressed proteins in the plasma of dairy cows between the inactive ovary and the normal estrus using the isobaric tags for relative and absolute quantitation (iTRAQ) proteomics. The same author subsequently screened out 135 significantly upregulated and 37 significantly downregulated proteins in the follicular fluid of estrus cows compared with non-estrus cows by the TMT-based proteomics [7]. Milk produced by dairy cows is a complex biofluid consisting of hundreds of different components that may be a good source for biomarkers discovery [8]. The variations of estrus status bring changes in the milk composition [4,9]. Furthermore, the readily available and non-invasive characteristics make milk preferable for biomarker discovery [10].

Relative few studies have been conducted regarding the estrus biomarker discovery in milk of dairy cows. In the present study, a TMT-based proteomics method was used to obtain an understanding of altered milk proteomic profiles of estrus in comparison with the non-estrus. The present study provides a new reference for discovering milk biomarkers, which may be helpful for further development of an easy and reliable estrus detection method for dairy cows.

## 2. Materials and Methods

### 2.1. Experimental Animals

Multiparous Holstein dairy cows with 3 to 5 parities on a commercial dairy farm in Hebei province, China, were used in this study. Cows were kept in free stall barns with cubicles equipped with a concrete solid floor and had free access to the feeding area. An ad libitum total mixed ration (TMR), consisting of corn silage, grass silage, concentrate, straw, and additives, were provided. Fresh TMR was supplied 3 times per day at 08:00, 16:00, and 24:00. The TMR was consistent during the experiment. Water was available continuously and ad libitum via troughs placed in the feeding area. The cows were milked three times daily at 07:00, 15:00, and 23:00 in a milk carousel with 80 milk stalls.

Cows at 15–21 d post-partum were synchronized with the following protocol: prostaglandin F_2α_ (PGF_2α_), 14 d later PGF_2α_, 7 d later gonadotropin-releasing hormone (GnRH), 7 d later PGF_2α_, 3 d later GnRH, 7 d later GnRH, 7 d later PGF_2α_, 24 h later PGF_2α_, 32 h later GnRH, and 14 h later timed artificial insemination (TAI). Pregnancy diagnosis was performed by an experienced veterinarian using both rectal palpation and ultrasound examination 5 weeks after TAI. To ensure that cows showed true estrus, only cows with insemination leading to conception were included. Cows confirming to be pregnant were considered to be in estrus at the day of insemination (day 0). Milk samples for proteomics analysis were collected at the day of insemination (day 0, the estrus group) and day −3 (3 days before the day of insemination, the non-estrus group) from 3 dairy cows confirming to be pregnant to examine possible biomarkers. The days in milk of those cows during collection ranged from 64 to 66 days.

### 2.2. Sample Collection

Milk samples were collected during the routine morning milking. Samples were taken into sterile tubes and transported to laboratory on ice within a few hours. All samples were subsequently stored at −80 °C until proteomics analysis.

### 2.3. TMT-Based Proteomics Analysis

#### 2.3.1. Protein Extraction

Milk samples were precipitated with four volumes of 10% trichloroacetic acid (TCA) for 4 h at −20 °C. After centrifugation at 4500× *g* for 5 min at 4 °C, the supernatant was removed. The remaining precipitate was washed with cold acetone for three times. The protein was dried and resuspended, and the protein content was determined with the BCA assay kit according to the manufacturer’s instructions (Beyotime Biotechnology, Shanghai, China).

#### 2.3.2. Trypsin Digestion

For digestion, the protein solution was reduced with 5 mM dithiothreitol for 30 min at 56 °C and alkylated with 11 mM iodoacetamide for 15 min at room temperature in darkness. The protein sample was then diluted by adding 100 mM TEAB to urea concentration less than 2 M. Finally, trypsin was added at 1:50 trypsin-to-protein mass ratio for the first digestion overnight and 1:100 trypsin-to-protein mass ratio for a second 4 h-digestion.

#### 2.3.3. TMT Labeling

After trypsin digestion, peptide was desalted by Strata X C18 SPE column (Phenomenex, Torrance, CA, USA) and vacuum-dried. Peptide was reconstituted in 0.5 mM TEAB and processed according to the manufacture’s protocol for TMT kit. Briefly, TMT reagent were thawed and reconstituted in acetonitrile. The peptide mixtures were then incubated for 2 h at room temperature and pooled, desalted and dried by vacuum centrifugation.

#### 2.3.4. HPLC Fractionation

The tryptic peptides were fractionated into fractions by high pH reverse-phase HPLC using Agilent 300Extend C18 column (Agilent Technologies Inc., Santa Clara, CA, USA, 5 μm particles, 4.6 mm ID, 250 mm length). Briefly, peptides were first separated with a gradient of 8% to 32% acetonitrile (pH 9.0) over 60 min into 60 fractions. Then, the peptides were combined into 6 fractions and dried by vacuum centrifuging.

#### 2.3.5. LC-MS/MS Analysis

The peptide segment was dissolved in liquid chromatography mobile phase A and separated using the EASY-nLC 1000 ultra high-performance liquid phase system (Thermo Fisher Scientific, Waltham, MA, USA). The mobile phase A was an aqueous solution containing 0.1% formic acid and 2% acetonitrile, and the mobile phase B contained 0.1% formic acid and 90% acetonitrile. The liquid phase gradient was set as follows: 0–40 min, 7–25% B; 40–52 min, 25–35% B; 52 to 56 min, 80% B, and maintain flow rate at 500.00 nL/min. The peptide segment was separated by a high-speed liquid phase system and injected into an NSI ion source for ionization before being analyzed by QE plus mass spectrometry. The ion source voltage was set to 2.1 kV, and the peptide parent ions and their secondary fragments were detected and analyzed using high-resolution Orbitrap. The scanning range of the primary mass spectrometry was set to 400–1500 *m*/*z*, and the scanning resolution was set to 70,000.00. The scanning range of the secondary mass spectrometry was fixed at a starting point of 100 *m*/*z*, and the resolution of the secondary scanning was set to 35,000.00. The data acquisition mode used the data dependency scanning program, that was, after the first level scanning, the parent ion of the first 20.00 peptide segment with the highest signal strength was selected to enter the HCD collision pool in turn to use 28% of the fragmentation energy for fragmentation, and the secondary mass spectrometry analysis was also carried out in turn. In order to improve the effective utilization of mass spectrometry, the automatic gain control was set to 5 × 10^4^ and the signal threshold was set to 3.8 × 10^4^ ions/s. The maximum injection time was set to 50 ms, and the dynamic exclusion time of tandem mass spectrometry scanning was set to 30 s to avoid repeated scanning of the parent ions.

#### 2.3.6. Database Search

The secondary mass spectrometry database was searched using Maxquant software (version 1.5.2.8). Retrieve parameters were set as follows: NCBI Bos taurus database, added a reverse library to calculate the false positive rate caused by random matching, and added common contamination libraries to the database to eliminate the impact of contaminated proteins in the identification results. The enzyme digestion method was set to Trypsin/P. The number of missing positions was set to 2. The minimum length of the peptide segment was set to 7 amino acid residues. The maximum modification number of peptide segments was set to 5. The tolerance for mass error of the primary parent ion in the first search and main search was set to 10.0 ppm and 5 ppm, respectively, and the tolerance for mass error of the secondary fragment ion was 0.02 Da. Carbamidomethyl on Cys, N-terminal TMT-6plex, and Lys TMT-6plex were specified as fixed modification and oxidation on Met were specified as variable modifications. FDR was adjusted to <1% and the minimum score for modified peptides was set to >40.

#### 2.3.7. Statistical Analysis

The identified proteins were analyzed by the non-parametric test of Wilcoxon sum-rank test using SPSS software (V18.0). Differentially expressed milk proteins between the estrus and non-estrus groups were determined using both fold change (fold change > 1.3 and fold change < 0.8) and statistical significance threshold (*p* < 0.05) based on the published reference [6,7].

#### 2.3.8. Bioinformatics Analysis of Differentially Expressed Proteins

This study used the DAVID online software (V6.8, https://david.ncifcrf.gov/) for GO analysis and KEGG pathway analysis. The *p*-value of GO and KEGG enrichment less than 0.05 was considered significant.

## 3. Results

### 3.1. Identification of Differentially Expressed Proteins

To analyze the milk proteome difference between estrus and non-estrus, the estrus group was compared with the non-estrus group. Based on the fold change > 1.3, fold change < 0.8, and *p* value < 0.05, a total of 89 milk proteins were found to be differentially expressed, of which 33 were upregulated and 56 were downregulated (Table 1).

### 3.2. GO Functional Classification of Differentially Expressed Proteins

To further extend the molecular characterization of the differentially expressed proteins, all differentially expressed proteins were subjected to GO analysis and categorized into biological process, cellular component, and molecular function based on their GO annotations (Figure 1). The results indicated that these proteins mainly belonged to the biological process of response to other organism, response to external biotic stimulus and response to biotic stimulus, the cellular component of extracellular region, and the molecular function of identical protein binding and lipid binding. As shown in Figure 1, the biological process of response to other organism, response to external biotic stimulus, and response to biotic stimulus had 15% protein involvement, respectively, followed by lipid localization, which enriched 9% protein. Lipid storage ranked the third at 6%. The most common cellular component was extracellular region, accounting for 55% of differentially expressed proteins. The second category was extracellular region part (47%), and the third was extracellular vesicle, extracellular organelle, and membrane-bounded vesicle (40%). Regarding classification based on molecular function, identical protein binding and lipid binding accounted for 8% of proteins and were the largest molecular functional category. GTP binding, guanyl ribonucleotide binding, and guanyl nucleotide binding followed at 7%. In addition, these differentially expressed milk proteins also showed high affinity with anion, protein complex, carboxylic acid, fatty acid, monocarboxylic acid, peptidoglycan, and long-chain fatty acid.

### 3.3. KEGG Pathway Enrichment of Differentially Expressed Proteins

The differentially expressed milk proteins were further evaluated using KEGG pathway analysis. This analysis indicated that the 89 differentially expressed proteins were significantly enriched in ten different pathways (Figure 2). The Neurotrophin signaling pathway included six proteins, followed by the Rap1 signaling pathway and the Ras signaling pathway which included five proteins, respectively. It was noted that the KEGG results showed significant enrichment in pathways of fatty acid biosynthesis and fatty acid metabolism, which were reported to be related to estrus regulation.

## 4. Discussion

In this study, the TMT-based quantitative proteomics approach was utilized to identify the differentially expressed proteins in milk samples between estrus and non-estrus dairy cows. Dairy cows which were synchronized and TAI were used. To ensure that cows showed true estrus, only cows with insemination leading to conception were included, and cows confirmed to be pregnant were considered to be in estrus at the day of insemination. A total of 89 proteins were significantly differentially expressed in the estrus group versus the non-estrus group, of which 33 were upregulated and 56 were downregulated based on the criteria of the fold change cut-offs of 1.3 and 0.8 and *p* value < 0.05. Ten differentially expressed proteins including smooth muscle alpha-actin 2 (ACTA2), perilipin 2 (PLIN2), matrix metalloproteinase 2 (MMP2), NAD(P)H steroid dehydrogenase-like (NSDHL), acetyl coenzyme A carboxylase α (ACACA), paraoxonase 1 (PON1), glycosylation-dependent cell adhesion molecule 1 (GLYCAM1), haptoglobin (HP), secretoglobin family 1D member (SCGB1D), and apolipoprotein B (APOB) identified in this study were in agreement with the findings of previous studies. For example, Forde et al. [11] reported that the expression of PLIN2 in the endometrium was modulated as the estrus cycle/early pregnancy progressed. ACTA2 was found to be differentially expressed in the anterior pituitary of heifers between the estrus and after ovulation via deep sequencing of the transcriptome [12]. APOB and PLIN2 were differentially expressed in milk of dairy cows between pregnancy and estrus [13]. Musavi et al. [14] observed that ACTA2, MMP9, NSDHL, and ACACA were differentially expressed in bovine endometrium between follicular stage and luteal stage. Zhao et al. [6] screened that PON1, GLYCAM1, HP, SCGB1D were differentially expressed in the blood of dairy cows between inactive ovary and the normal estrus. The concentration of HP in blood at estrus phase was significantly higher than diestrus and proestrus in dairy cows [15].

Bioinformatics analysis in the current study revealed that PLIN2, PON1, APOB, and NSDHL participated in lipid binding, lipid storage, lipid localization, and lipid metabolic process. Lipid is a vital source of energy and serves as the main substrate in oocytes [16,17]. During oocyte maturation, active lipid synthesis, transport, storage, and degradation occur in the oocyte and surrounding cumulus cells. Lipid metabolism affects oocyte maturation and therefore has close correlation with estrus regulation. It was reported that the onset of ovarian activity after calving could be affected by the amount of body lipid and the body lipid change. Body lipid mobilization during the postpartum period increased estrus duration and the number of standings to be mounted per estrus [18].

PLIN2 is a cytosolic protein that prevents lipid degradation and promotes lipid accumulation [19,20]. PLIN2 has been shown to be upregulated in oocytes of preovulatory follicles, suggesting its potential role in oocyte meiotic resumption [21]. PON1, a calcium-dependent serum enzyme, is involved in lipid metabolism. The activity of PON1 is associated with follicular development, ovulation, and ovarian function of postpartum in dairy cows [22]. The activity of PON1 in follicular fluid was significantly higher in healthy active follicles than in atretic follicles of dairy cows [23]. In a previous study about postpartum dairy cows, those with greater PON1 activity at 7 days postpartum had the ovulation earlier than cows with less PON1 activity at 7 days postpartum. APOB is the critical structural component of lipoproteins. The APOB-containing lipoproteins contain large amounts of cholesterol and triglycerides and are involved in lipoprotein metabolism and lipid transport. The pre-implantation embryo needs cholesterol for membrane formation during its cell cleavages, and triglycerides act as an energy source during oocyte maturation and potential during pre-implantation embryo development [24]. Therefore, APOB has a positive and significant effect on oocyte retrieval, oocyte fertilization, and embryo quality. Moreover, the expression and content of APOB for luteinized granulosa cells cultured in the presence of gonadotropins was significant reduced [25]. NSDHL, a sterol dehydrogenase which is related to cholesterol synthesis, is responsible for the lipid deposition. Yang et al. [26] has revealed that cholesterol synthesis is closely associated with ovarian reserve function, whereas patients with diminished ovarian reserve exhibit significantly lower cholesterol metabolic levels compared with those with normal ovarian reserve. These evidences indicate that PLIN2, PON1, APOB, and NSDHL may play an important role in estrus regulation.

In oocyte, fatty acids are stored in lipid droplets as triglycerides. Free fatty acids obtained from triacylglyceroles can be directly transported from lipid droplets to mitochondria and metabolized by β-oxidation, which reveals a fast and direct way of energy acquisition for the oocyte [27,28]. In this study, ACACA was found to be enriched in fatty acid biosynthesis and metabolism by bioinformatics analysis. Indeed, ACACA is a biotin-dependent enzyme [29], and the C-terminal carboxytransferase domain of ACACA catalyzes the carboxylation of acetyl-CoA to malonyl-CoA and, therefore, is involved in the fatty acid synthesis [30]. Thus, ACACA may also play an important role in estrus regulation.

Combined with the above evidence, we propose that ACACA, APOB, NSDHL, PLIN2, and PON1 might serve as potential biomarkers in milk to detect dairy cows at estrus. Moreover, in the present study, milk samples were collected from dairy cows which were synchronized and timed artificial insemination (TAI). Nowadays, synchronization protocols that allow for TAI are widespread adoption in the dairy industry. However, in practice, TAI has been challenged due to unsatisfactory ovulation rates and low pregnancy rates. The main problem is that not all cows ovulate within a fixed period after injecting various hormones in the TAI procedure [31]. Although estrus detection is not required with TAI, several studies have shown that regardless of protocol used, expression of estrus before TAI can increase pregnancy rates [32,33]. Therefore, altering TAI protocols to improve the proportion of dairy cows in estrus before TAI is needed. Submitting only animals in estrus to TAI, and delaying insemination in the remaining animals to allow for the display of estrus before AI, results in improved estrus rates. Therefore, optimizing the timing of insemination is important for pregnancy [34]. The cows confirming to be pregnant may in peri-estrus or peri-ovulation and have a fertile oocyte before AI, and these potential milk biomarkers identified in the present study may be used for diagnoses of dairy cows in estrus or, even better, those approaching estrus, subsequently guiding their optimal timing of AI.

## 5. Conclusions

In conclusion, the present study characterized the milk proteome of dairy cows and identified 89 differentially expressed proteins between estrus and non-estrus dairy cows. Among these proteins, ACACA, APOB, PLIN2, NSDHL, and PON1 were found to be enriched in lipid metabolism and fatty acid metabolism, which have a close correlation with estrus regulation. These five proposed milk proteins may act as potential indicators of estrus, which deserve further study. Although the sample size was low, which was the limitation of the present study, this work can be considered a preliminary study, and the current results can provide a reference for the further study in this specific subject. In future, more milk samples should be collected from the dairy cows of estrus and non-estrus to further validate the expression patterns of these five milk proteins. Moreover, the mechanisms of these proteins to regulate the estrus processes and their potential value in estrus detection should also be demonstrated in further research.

## Figures and Tables

**Figure 1 animals-13-02892-f001:**
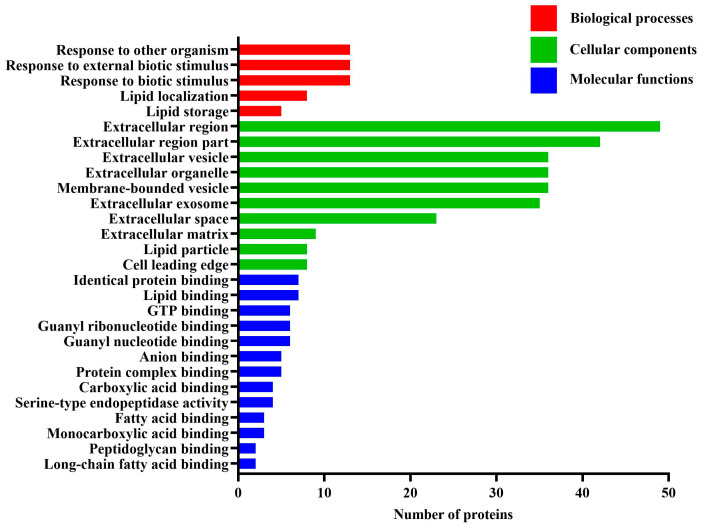
Gene ontology annotation of the differentially expressed milk proteins between the estrus and non-estrus dairy cows.

**Figure 2 animals-13-02892-f002:**
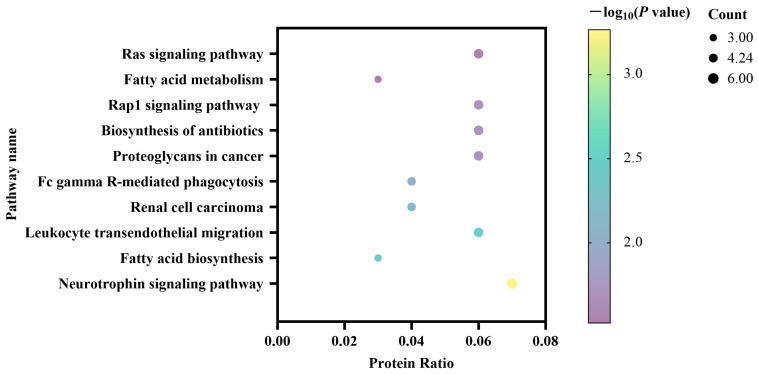
Kyoto Encyclopedia of Genes and Genomes pathways of the differentially expressed milk proteins between the estrus and non-estrus dairy cows.

**Table 1 animals-13-02892-t001:** Differentially expressed milk proteins between the estrus and non-estrus dairy cows.

Protein Accession	Protein Description	Gene Name	MFC	U/D	*p* Value
F1MTB5	F-box and leucine rich repeat protein 19	*FBXL19*	0.72	D	0.00
E1BBV1	Round spermatid basic protein 1	*RSBN1*	2.78	U	0.00
A0A3Q1MJ34	40S ribosomal protein S9	*RPS9*	3.09	U	0.02
F1N1Z8	Uncharacterized protein	*LOC104975830*	0.66	D	0.00
Q3ZC35	Cellular communication network factor 1	*CCN1*	0.66	D	0.04
P61223	Ras-related protein Rap-1b	*RAP1B*	0.77	D	0.01
Q3T0Q4	Nucleoside diphosphate kinase B	*NME2*	0.71	D	0.00
Q0VD31	F-box/LRR-repeat protein 4	*FBXL4*	1.93	U	0.00
P62157	Calmodulin	*CALM*	0.56	D	0.00
Q58CP2	Inactive C-alpha-formylglycine-generating enzyme 2	*SUMF2*	1.39	U	0.03
Q2KJ93	Cell division control protein 42 homolog	*CDC42*	1.32	U	0.03
F1MGR1	Sodium/nucleoside cotransporter	*SLC28A3*	1.43	U	0.00
A0A3Q1M483	Acetyl-CoA carboxylase 1	*ACACA*	1.49	U	0.00
A0A3Q1M7Y8	Procollagen-lysine,2-oxoglutarate 5-dioxygenase 1	*PLOD1*	0.72	D	0.03
A0A3Q1NIJ7	Heparanase	*HPSE*	0.64	D	0.00
A0A3Q1N0L4	Dynamin binding protein	*DNMBP*	0.69	D	0.03
A0A3Q1LXG9	von Willebrand factor A domain-containing protein 1	*VWA1*	0.65	D	0.01
A0A3Q1LSP4	Complement C8 alpha chain	*C8A*	0.73	D	0.01
A0A3Q1MAQ8	Serine protease 8	*PRSS8*	1.39	U	0.02
G3N0M6	CIDE-N domain-containing protein	*CIDEA*	1.61	U	0.00
E1BQ28	Solute carrier family 39 member 8	*SLC39A8*	1.52	U	0.00
E1BNE1	Lipid droplet associated hydrolase	*LDAH*	1.32	U	0.04
Q2KIS4	Dehydrogenase/reductase (SDR family) member 1	*DHRS1*	1.35	U	0.00
E1BJF9	Serum amyloid A protein	*LOC104968478*	1.42	U	0.02
G3N2S4	Leucine rich repeat containing 26	*LRRC26*	0.65	D	0.04
F1MXQ3	FAM20C golgi associated secretory pathway kinase	*FAM20C*	0.76	D	0.00
F1MUL0	Leukocyte surface antigen CD47	*CD47*	1.35	U	0.01
F1MYL3	Syntaxin-binding protein 6	*STXBP6*	1.40	U	0.02
E1BJ18	Calpain 6	*CAPN6*	1.32	U	0.00
F1MEX9	Acyl-CoA synthetase long chain family member 3	*ACSL3*	1.34	U	0.00
F1MXP8	Prosaposin	*PSAP*	0.76	D	0.00
A8E654	COL18A1 protein	*COL18A1*	0.75	D	0.00
Q2KIW1	Paraoxonase 1	*PON1*	0.66	D	0.03
A6QPW7	TNF receptor superfamily member 6b	*TNFRSF6B*	0.63	D	0.00
Q0VCZ8	Acyl-CoA synthetase long-chain family member 1	*ACSL1*	1.41	U	0.00
Q0IIA2	Odorant-binding protein-like	*MGC151921*	0.68	D	0.02
F1MN49	Eukaryotic translation initiation factor 5A	*EIF5A2*	1.31	U	0.03
A0A3Q1MUW9	F-box/LRR-repeat protein 20	*FBXL20*	1.59	U	0.01
F6Q8A8	Nucleobindin 2	*NUCB2*	0.60	D	0.00
A0A3Q1MLF5	AP complex subunit sigma	*AP1S2*	0.73	D	0.04
Q3SYR8	Immunoglobulin J chain	*JCHAIN*	0.70	D	0.04
A2VDV1	Ankyrin repeat domain 22	*ANKRD22*	1.51	U	0.03
A8E4N5	RING1 protein	*RING1*	0.71	D	0.04
A6QPK0	SCGB2A2 protein	*SCGB2A2*	0.73	D	0.02
Q32KV6	Nucleotide exchange factor SIL1	*SIL1*	0.61	D	0.00
Q0VCP3	Olfactomedin-like protein 3	*OLFML3*	1.96	U	0.00
Q3ZBE9	Sterol-4-alpha-carboxylate 3-dehydrogenase, decarboxylating	*NSDHL*	1.48	U	0.00
P62998	Ras-related C3 botulinum toxin substrate 1	*RAC1*	1.36	U	0.00
Q9TUM6	Perilipin-2	*PLIN2*	1.46	U	0.00
P62833	Ras-related protein Rap-1A	*RAP1A*	1.51	U	0.01
F1N152	Serine protease HTRA1	*HTRA1*	0.75	D	0.00
P19803	Rho GDP-dissociation inhibitor 1	*ARHGDIA*	1.30	U	0.00
P80195	Glycosylation-dependent cell adhesion molecule 1	*GLYCAM1*	0.69	D	0.00
P42916	Collectin-43	*CL43*	0.73	D	0.01
P10790	Fatty acid-binding protein, heart	*FABP3*	1.35	U	0.00
P18892	Butyrophilin subfamily 1 member A1	*BTN1A1*	1.38	U	0.02
P81134	Renin receptor	*ATP6AP2*	0.74	D	0.00
O46375	Transthyretin	*TTR*	0.76	D	0.00
A0JNP2	Secretoglobin family 1D member	*SCGB1D*	0.74	D	0.00
Q0IIG8	Ras-related protein Rab-18	*RAB18*	1.32	U	0.00
Q3SZF2	ADP-ribosylation factor 4	*ARF4*	1.49	U	0.02
G3X807	Histone H4	*H4C9*	0.46	D	0.00
G3MZ19	Jacalin-type lectin domain-containing protein	*ZG16B*	0.66	D	0.01
F1MMS7	SERPIN domain-containing protein	*/*	0.43	D	0.01
P56425	Cathelicidin-7	*CATHL7*	0.66	D	0.04
A0A3Q1MFR4	Apolipoprotein B	*APOB*	1.31	U	0.00
E1B8Q6	Uncharacterized protein	*LOC112441458*	0.59	D	0.02
F6S1Q0	Keratin 18	*KRT18*	0.41	D	0.01
A0A3Q1MKJ8	Syndecan	*SDC2*	0.75	D	0.01
A0A452DJE4	Actin, aortic smooth muscle	*ACTA2*	0.66	D	0.01
E1B6Z6	Lipocalin 2	*LCN2*	0.53	D	0.00
A5D984	Pyruvate kinase	*PKM*	0.69	D	0.00
A8E4P3	STOM protein	*STOM*	0.72	D	0.02
A0A3Q1MB98	Haptoglobin	*HP*	0.59	D	0.00
Q9XSJ4	Alpha-enolase	*ENO1*	0.68	D	0.00
O02853	Prostaglandin-H2 D-isomerase	*PTGDS*	0.71	D	0.00
P27479	Arachidonate 15-lipoxygenase	*ALOX15*	1.32	U	0.01
P52176	Matrix metalloproteinase-9	*MMP9*	0.52	D	0.01
Q5E9F7	Cofilin-1	*CFL1*	0.76	D	0.00
P22226	Cathelicidin-1	*CATHL1*	0.58	D	0.00
Q3MHR7	Actin-related protein 2/3 complex subunit 2	*ARPC2*	0.52	D	0.02
Q1JPB0	Leukocyte elastase inhibitor	*SERPINB1*	0.51	D	0.02
P33046	Cathelicidin-4	*CATHL4*	0.63	D	0.02
P19660	Cathelicidin-2	*CATHL2*	0.60	D	0.01
Q3T149	Heat shock protein beta-1	*HSPB1*	0.70	D	0.02
Q8SPP7	Peptidoglycan recognition protein 1	*PGLYRP1*	0.60	D	0.04
P28782	Protein S100-A8	*S100A8*	0.39	D	0.00
Q5E956	Triosephosphate isomerase	*TPII*	0.59	D	0.03
P48616	Vimentin	*VIM*	0.70	D	0.00

MFC: mean fold change; U: upregulated expressed proteins; D: downregulated expressed proteins.

## Data Availability

The data that support the findings of this study are available from the corresponding authors upon reasonable request.

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
