# Peer review of "Differences in Milk Proteomic Profiles between Estrous and Non-Estrous Dairy Cows"

_animals, 2023, doi:10.3390/ani13182892_

Round 1

Reviewer 1 Report

The study by Du et al., although not entirely novel, is well executed although a higher number of estrus and not-estrus animals could have been involved, in order to detect UP and DOWN regulated genes of interest. The authors are missing though the point to give informations (DISCUSSION) in the paper on the possible costs of such procedure with regard to identification of estrus. This could also be put in line with the handling of this procedure and the comparison with other reproductive approaches that do not use the detection of estrus, but rather rely on fixed timed AI. They conclude that of the 89 differentially expressed proteins, 5 milk proteins could be used as potential indicator of estrus, although their role in the regulation of estrus is still not clear. This study would reach a higher level of importance if those differentially expressed proteins could be pu to test. Finally, authors do not give indication on the timing of such approach., i.e. when using it? At random among animals in the course of their cycle? Would it be better used in order to confirm the right timing of estrus in the course of synchronization protocols? All this would enrich the Discussion and possibly reach the likelywood conclusion on the best use of differentially expressed milk proteins with regard to estrus detection and synchronization protocols for AI

a moderate revision of written Englisg is required

Reviewer 2 Report

General comments

This study aimed to identify potential biomarkers in milk to detect dairy cows at estrus. The sample size is very low, and, apparently, milk sample collections were taken in the same dairy cows in two different moments. It is necessary that the authors report what non-parametric test was used to compare both moments, and to confirm that is adequate for this study design and sample size. The statistical test needs to be adequate for repeated measures. Also, a clarification regarding the fold change cut-offs should be done. Despite these issues, the manuscript is well written and presented (including Table 2 and figures), and the findings sufficiently discussed according to the nature of the results. In my opinion, the information reported by Table 1 (Day of TAI after calving, and parity of cows) can be done in text. The limitations of the study should be emphasized (I also suggest a mention in the conclusion section as reported in the abstract). A minimum of 5 cows would give more support to the findings, certainly. Nevertheless, I understand that the current information can serve to delineate further studies in this specific subject. Nonetheless, the results should be interpreted with extreme caution due to the very low sample size. In my opinion, this work should be addressed as a preliminary study. The references are adequate regarding the content of the manuscript.

Specific comments

L58-59: Please use the term “estrus” instaed of “oestrous” to identify estrus period. Check the manuscript.

L89-91: You report that milk samples were collected in the same cows at different periods.  So, you don´t have two groups but only one group.

L157: Please, identify the non-parametric test.

L168-169: Please clarify the the fold change cut-offs and it relationschips with MFC (L172). Why these thresholds were used (no reference was added)?
